# Ranolazine Interacts Antagonistically with Some Classical Antiepileptic Drugs—An Isobolographic Analysis

**DOI:** 10.3390/molecules27248955

**Published:** 2022-12-15

**Authors:** Kinga Borowicz-Reutt, Monika Banach

**Affiliations:** Independent Unit of Experimental Neuropathophysiology, Department of Toxicology, Medical University of Lublin, Jaczewskiego 8b, PL-20-090 Lublin, Poland

**Keywords:** ranolazine, first-generation antiepileptic drugs, maximal electroshock, interactions, isobolographic analysis, delayed lethality

## Abstract

Ranolazine, an antianginal and antiarrhythmic drug blocking slow inactivating persistent sodium currents, is described as a compound with anticonvulsant potential. Since arrhythmia often accompanies seizures, patients suffering from epilepsy are frequently co-treated with antiepileptic and antiarrhythmic drugs. The aim of this study was to evaluate the effect of ranolazine on maximal-electroshock (MES)-induced seizures in mice as well as interactions between ranolazine and classical antiepileptic drugs in this model of epilepsy. Types of pharmacodynamic interactions were established by isobolographic analysis of obtained data. The main findings of the study were that ranolazine behaves like an antiseizure drug in the MES test. Moreover, ranolazine interacted antagonistically with carbamazepine, phenytoin, and phenobarbital in the proportions of 1:3 and 1:1. These interactions occurred pharmacodynamic, since ranolazine did not change the brain levels of antiepileptic drugs measured in the fluorescence polarization immunoassay. Ranolazine and its combinations with carbamazepine, phenytoin, and phenobarbital did not impair motor coordination evaluated in the chimney test. Unfortunately, an attempt to conduct a passive avoidance task (evaluating long-term memory) resulted in ranolazine-induced delayed lethality. In conclusion, ranolazine exhibits clear-cut anticonvulsant properties in the MES test but interacts antagonistically with some antiepileptic drugs. The obtained results need confirmation in clinical studies. The mechanisms of ranolazine-induced toxicity require specific explanation.

## 1. Introduction

Ranolazine is a piperazine derivative increasingly used as an antianginal and antiarrhythmic drug. Interestingly, its molecular structure resembles that of lidocaine and mexiletine, representatives of class IB antiarrhythmic drugs [Figure 1]. Both lidocaine and mexiletine present established anticonvulsant properties determined in either experimental or clinical investigations [1,2].

Growing evidence indicates the usefulness of ranolazine in the treatment and prevention of supraventricular and ventricular cardiac arrhythmias, particularly atrial fibrillation in patients with acute coronary syndromes and after percutaneous transluminal cardiovascular angioplasty. There are also reports that ranolazine may be effective in the treatment of pulmonary hypertension, congenital myotonia, and diabetic neuropathic pain [3]. The drug reduced also cramps in patients with amyotrophic lateral sclerosis [4]. 

The mechanism of action of ranolazine seems to be quite complex. Firstly, it acts as a multichannel blocker such as, for example, amiodarone, a class III antiarrhythmic drug. Ranolazine primarily blocks inward (depolarizing) sodium and calcium L type currents, as well as repolarizing potassium IKr and IKs currents. The drug weakly antagonizes α1-adrenergic receptors in blood vessels and β1-adrenergic receptor in the heart muscle. The metabolic effect of ranolazine is also emphasized. This antianginal agent reduces glucose plasma levels and increases glucose metabolism in the heart muscle, raising cAMP production. Surprisingly, ranolazine can also be used in oncology. It was proved that cancer cells that express persistent sodium channels are more metastatic. In experiments conducted in vitro and in mice, the drug decreased the invasiveness of human breast cancer cells [5]. In addition, ranolazine presented anti-inflammatory action, decreasing CRP plasma levels in patients with ischemic heart disease. In experimental conditions, the drug reduced progression of atherosclerotic plaques development in LDL receptor knockout mice. This effect was related to inhibition of the NFkB pathway and, in consequence, reduced production of adhesion molecules and proinflammatory cytokines in endothelial cells [6]. 

Referring to sodium channels, ranolazine inhibits both fast (I_NaF_) and late (persistent) inactivating (I_NaL_) sodium currents. The action of this drug is, however, mostly attributed to the use-dependent blockade of I_NaL_s activated in the atria in pathological conditions. Increased late sodium currents are sufficient to increase the intracellular concentration of sodium, and, in consequence, calcium ions. Calcium overload disturbs the electrophysiological homeostasis in cardiomyocytes and leads to arrhythmias and decreased heart contractility [1]. 

Importantly, ranolazine was reported to block sodium currents not only in the heart muscle, but also in some lines of cultured neurons. In fact, the drug is able to inhibit some sodium channel isoforms in cardiomyocytes (NaV1.5), skeletal muscles (NaV1.4), peripheral neurons (NaV1.7 and NaV1.8), and brain cells (NaV1.1). Mutations in the SCN1A gene coding the alpha subunit of NaV1.1 are associated with generalized epilepsy with febrile seizures, Dravet syndrome (severe myoclonic epilepsy in infancy), and inherited familial migraine. Simultaneously, such mutations resulted in an increase in persistent sodium currents [7]. Moreover, ranolazine reduced the epileptiform activity induced by glutamatergic N-methyl D-aspartate (NMDA) activation in rat hippocampal neurons. However, the drug did not significantly affect the voltage-gated potassium channels or NMDA- and GABA-related neurotransmissions. Additionally, the neurotransmitter release to the synaptic cleft was not influenced [8]. In preclinical investigations, ranolazine and GS967, a more potent inhibitor of I_NaL_ currents, reduced seizure frequency in *SCN2aQ54* mice by 50 and 90%, respectively. The genetically modified mouse line *SCN2aQ54* with inactivated NaV1.2 channels presents partial seizures followed by generalized convulsions. In addition, GS967 protected mice against maximal-electroshock-induced seizures [9]. This suggests an important role of the I_NaL_ current inhibitors in epileptogenesis and opens up the possibilities of their use in the treatment of some types of epilepsy. 

It is often underlined that ranolazine is devoid of proarrhythmogenic effects. In clinical studies, the drug was well tolerated up to a dose of 2000 mg/day [4,5]. At recommended doses, approximately 6% of patients discontinued treatment with ranolazine due to adverse effects. For comparison, approximately 3% of subjects on placebo quitted trial because of undesired effects. The most common reasons were dizziness, headache, constipation, nausea, vomiting, and asthenia. In humans, ranolazine is metabolized by CYP3A4, and partially CYP2B6 [5,9]. Therefore, inhibitors of the two isoenzymes, including fluconazole, diltiazem, and macrolides, can increase plasma concentrations and toxicity of ranolazine.

Due to frequent co-occurrence of arrhythmias and seizures, as well as a theoretical basis of the anticonvulsant action of ranolazine, we decided to determine interactions between this antianginal medication and classical antiepileptic drugs in the maximal electroshock test in mice. Maximal electroshock is a standard screening method for the identification of potential anticonvulsant drugs. It has been used for years due to the ease and low costs of performing. For the same reasons, this model is attractive for carrying out isobolographic analysis of drug interactions [10].

## 2. Results

### 2.1. Interactions between Ranolazine and Classical Antiepileptic Drugs in Maximal-Electroshock-Induced Seizures in Mice—An Isobolographic Analysis

Values of ED_50_ (50% effective doses) calculated for ranolazine and antiepileptic drugs on the basis of results obtained in the MES test are presented in Table 1. The isobolographic analysis showed antagonistic interaction between ranolazine and carbamazepine, ranolazine and phenytoin, as well as ranolazine and phenobarbital in the two fixed dose-ratios of 1:3 and 1:1. Additivity was found between ranolazine and three above-mentioned antiepileptic drugs applied in a proportion of 3:1. Similarly, the additive interaction was observed between ranolazine and valproate administered in all three fixed dose-ratios (Table 2, Figure 1). 

### 2.2. Effect of Ranolazine, Antiepileptic Drugs, and Combinations of Ranolazine with Antiepileptic Drugs on Motor Coordination in Mice

Ranolazine and classical antiepileptic drugs administered alone at their ED_50_s as well as combinations of ranolazine and respective antiepileptic drugs applied at all fixed dose-ratios did not produce significant motor impairment evaluated in the chimney test (Table 3).

### 2.3. Influence of Ranolazine on the Brain Concentrations of Antiepileptic Drugs

Ranolazine was combined with antiepileptic drugs in two proportions (1:3 and 1:1), in which antagonism was observed between the antianginal drug and carbamazepine, phenytoin and phenobarbital. 

For valproate, the same proportions were used to maintain comparability of results. In the 1:3 proportion, ranolazine significantly diminished the brain concentration of valproate from 78.71 ± 4.73 to 84.01 ± 5.71 µg/mL, indicating pharmacokinetic interaction. In all remaining determinations, ranolazine did not change the brain levels of antiepileptic drugs, suggesting that the interactions found seem to be pharmacodynamic (Figure 2).

## 3. Discussion

Results of the present study revealed that ranolazine behaved like a regular antiepileptic drug in the maximal electroshock test in mice. Isobolographic analysis of obtained data showed antagonistic interactions between ranolazine and carbamazepine, ranolazine and phenytoin, as well as ranolazine and phenobarbital, when drugs were applied in the two fixed dose-ratios of 1:3 and 1:1. Combinations of ranolazine and valproate at these proportions led to additive interaction, however, a tendency to antagonism was clearly observed. Probably, in the ratio of 1:3, antagonism would be observed if the antianginal drug did not increase the brain level of valproate. Interestingly, ranolazine interacted additively with all four antiepileptics in the proportion of 3:1. It can be assumed that antagonism was manifested in combinations in which lower doses of ranolazine were used. Worth mentioning is that the only pharmacokinetic interaction was showed between ranolazine and valproate in the fixed ratio of 1:3. Remaining interactions seem to be pharmacodynamic. 

Since ranolazine inhibits primarily persistent sodium currents (I_NaL_), the anti-electroshock effect of ranolazine could be most likely attributed to this mechanism of action. Fast inactivated sodium currents (I_NaF_) are also blocked by ranolazine, however, to a much lesser extent. It is widely known that I_NaF_ currents are the main target for numerous antiepileptic drugs, including phenytoin, carbamazepine, valproate, and lamotrigine. These medications decrease neuronal excitability and repetitive firing by stabilizing the inactivated state of sodium channels. Nevertheless, persistent sodium currents are also thought to be involved in the generation and propagation of action potential, as well as to support repetitive neuronal firing. Among the first-generation antiepileptic drugs, phenytoin inhibits not only I_NaF_, but also I_NaL_ currents [7]. Moreover, lacosamide, one of the third-generation antiepileptics, reduces mostly I_NaL_, minimally affecting I_NaF_ currents [8]. In silico study evidenced that persistent sodium current blockers suppress seizures caused by mutation in subfamily A of voltage-gated potassium channels (K_v_1) [11]. In general, voltage-gated potassium channels are involved in the moderation of neuronal excitability and pathogenesis of some types of epilepsies [12]. It is worth emphasizing that these channels are not directly affected by ranolazine [8]. 

Also blocking L-type calcium channels can contribute to the antiseizure action of ranolazine. Numerous antagonists of these channels presented antiseizure action in both clinical and preclinical studies. For instance, verapamil showed promising therapeutic effect in patients with status epilepticus and drug-resistant epilepsies, such as Dravet or Lennox–Gastaut syndromes [13]. Nicardipine attenuated development of pentylenetetrazole-induced kindled seizures in rats [14]. Amlodipine presented anticonvulsant action in pentylenetetrazole- and maximal-electroshock-induced seizures in mice [15]. Detailed information on the anticonvulsant effects of L-type calcium channel blockers, including amlodipine, nimodipine, nifedipine, niguldipine, isradipine, verapamil, and diltiazem, has been collected in the review article of Kułak et al. [16]. 

In contrast, blocking the delayed rectifier potassium currents (I_K_s) seems to counteract the antiseizure action of ranolazine. Notably, I_K_s consist of rapid (I_Kr_) and slow (I_Ks_) components that differ in kinetic parameters and ligand sensitivity [17]. Mutations of IKs channels were reported to induce seizures related to long QT syndrome [18]. Dysfunctions of IKr are thought to be involved in epileptogenesis [19]. On the other hand, drugs inducing QT prolongation, thus increasing the risk of arrhythmias and seizures, were reported to block the ‘rapid’ cardiac delayed rectifier potassium currents (IKrs) [20]. 

Interestingly, phenytoin, phenobarbital, and to a much lesser extent, carbamazepine, were shown to inhibit IKr currents. According to authors, this could increase the risk for sudden unexpected death in patients with epilepsy treated with these drugs [21]. There are no available data on the action of valproate on rectifying potassium currents. 

Ranolazine exhibited not only anticonvulsant, but also proconvulsant effects in experimental animals. For instance, in carbamazepine-resistant rats developed in the model of window-pentylenetetrazole kindling. The authors tried to explain this phenomenon by assumption that the role of I_NaL_ currents in generation and propagation of seizures in this model may be less important than in other types of experimental seizures [22]. 

As it was mentioned in the Introduction, ranolazine structurally resembles lidocaine and mexiletine (both from class Ib antiarrhythmic drugs), while its mechanism of action is to some extent similar to amiodarone, a representative of class III antiarrhythmics. Lidocaine and mexiletine presented anticonvulsant action in some experimental seizure models [2]. However, lidocaine is better known for its proconvulsant action, especially when applied at higher doses (70–90 mg/kg). This effect was described to be similar to that induced by bicuculline [23]. There are no data about interactions between lidocaine and antiepileptic drugs. However, lidocaine and carbamazepine were reported to bind to similar sites within voltage-gated sodium channels, indicating the possibility of simple additive or antagonistic, but not synergistic interactions [24]. In contrast, lidocaine and phenytoin bind to different sites of the sodium channel, giving a background to the synergistic interaction. Similarly, phenobarbital acting primarily on GABA_A_ receptors can interact synergistically with lidocaine [25]. However, none of the above-mentioned theoretical considerations were proved in experimental conditions. Mexiletine, similarly to ranolazine, exhibited properties of an antiseizure drug in the maximal electroshock test in mice. Isobolographic analysis revealed an antagonistic interaction between mexiletine and valproate in two fixed-ratio combinations of 1:1 and 3:1. However, this effect could be due partially to the ranolazine-induced decrease in the brain concentration of valproate. Additivity was observed between mexiletine and valproate in the proportion of 1:3. Moreover, this antiarrhythmic drug interacted additively with carbamazepine, phenytoin, and phenobarbital in all three fixed ratios, which was not related to altered brain concentrations of antiepileptic drugs [2]. Finally, amiodarone, a representative of class III antiarrhythmics without its own antiseizure action, significantly enhanced the anti-electroshock activity of carbamazepine. Effects of valproate, phenytoin, and phenobarbital were not affected. Furthermore, amiodarone did not influence the brain concentrations of antiepileptic drugs [2]. Summing up, the interactions of ranolazine with classical antiepileptic drugs seem to be unique and differ from those observed with antiarrhythmic drugs with a similar structure or mechanism of action.

At present, we do not have sufficient data to determine the causes of antagonistic interactions between ranolazine and carbamazepine, phenytoin, or phenobarbital. It used to be thought that synergistic interaction occurs between drugs with complementary mechanisms of action, whereas antagonism may be observed more often between compounds with overlapping mechanisms [26]. Following this line of reasoning, antagonism demonstrated in this study could be attributed to the concomitant blocking of IKr currents by ranolazine and antiepileptic drugs. Also blocking L-type calcium channels could contribute to such an interaction, at least in the case of ranolazine and carbamazepine or phenytoin. However, the affinity of these medications to calcium channels is very weak [25,27]. 

As regards undesired effects, ranolazine and antiepileptic drugs administered at their ED_50_ doses did not impair motor coordination. Similarly, no motor disturbances were observed in mice administered with combinations of ranolazine and antiepileptic medications given in proportions of 1:3 and 1:1. We planned to also investigate the influence of ranolazine and its combinations on long-term memory in mice. Unfortunately, it was revealed to be impossible because of the delayed toxicity of ranolazine. In the test of passive avoidance, evaluating cognitive processes, experiments are continued 24 h after drug administration. However, no mice survived in the group injected with ranolazine at the ED_50_ dose (96.4 mg/kg), which corresponds to 547 mg in humans [28]. In groups applied with drug combinations, in which doses of ranolazine were lower, up to 6 of 10 mice died within 24 h. The delayed mortality made the obtained results not reliable. For now, we cannot explain this unfavorable phenomenon, especially given that during chronic treatment in humans ranolazine up to 2000 mg/day was very well tolerated. The same dose was established as a maximum recommended daily dose of this drug [4,5]. In rats, reported LD_50_ (50% lethal dose) of ranolazine administered orally was 980 mg/kg [29]. Unfortunately, there are no data on LD_50_ in mice. In clinical conditions, only three cases of intoxication with ranolazine were reported so far, but all of them resulted from intentional overdose [30]. In the current state of knowledge, we are not able to determine potential causes of ranolazine-induced delayed lethality. It can only be suggested that ranolazine-related inhibition of IKr and IKs currents may contribute to this phenomenon in the mechanism of inducing severe arrhythmias and/or seizures [18,19,20].

## 4. Materials and Methods

### 4.1. Animals

All experiments were carried out on 20–25 g male Albino Swiss mice weighing 20–25 g. Mice were admitted for testing by the Animal Welfare Committee. The rodents were kept in colony cages with provided standard conditions of space, temperature, humidity, air exchange, natural day-night cycle 12/12 h, and constant access to water and food. After 7-day acclimatization, animals were randomly selected to experimental groups counting 8–10 subjects. All procedures were conducted between 9 a.m. and 3 p.m. Methodology of experiments was approved by the Local Ethical Committee at the University of Life Sciences in Lublin (license No KE 42/2015) as well as met the requirements of AR290 RIVE guidelines and EU Directive 2010/63/EU for animal experiments.

### 4.2. Drugs

An antianginal agent ranolazine (RNL; Ranexa, Menarini International Operations Luxembourg S.A., Luxembourg) and classical antiepileptic drugs: valproate (VPA), carbamazepine (CBZ), phenytoin (PHT), and phenobarbital (PB) were used in the study. VPA, CBZ, and PHT were purchased from Sigma (St. Louis, MO, USA). PB was obtained from UNIA Pharmaceutical Department (Warsaw, Poland). All substances used in the study were suspended in a 1% aqueous solution of Tween 80 (Sigma-Aldrich, St. Louis, MO, USA). Subsequently, drug suspensions were injected intraperitoneally (ip) in a volume of 10 mL/kg body weight at times corresponding to their maximal anti-electroshock action in mice. RNL was injected 15 min prior to tests. Among antiepileptic drugs, VPA and CBZ were applied 30 min, PB—60 min, whereas PHT—120 min before procedures. 

### 4.3. Maximal Electroshock Seizure Test

The test of maximal-electroshock-induced seizures (MES) in rodents is classified as a model of generalized tonic–clonic convulsions. It is commonly employed in screening tests of new potential anticonvulsants [10]. The anticonvulsant effects of ranolazine, antiepileptic drugs, and their combinations were determined in the MES test and expressed as ED_50_s, median effective dose protecting 50% of mice against tonic convulsions. To determine the ED_50_ of a given drug, 3–5 groups of mice were administered with increasing doses of this drug and subjected to the maximal electroshock by ear-clip electrodes. Subsequently, a dose–effect relationship was designated on the basis of the percentage of mice protected against seizures according to Litchfield and Wilcoxon [31]. The current parameters in the MES were: intensity of 25 mA, frequency of 50 Hz, voltage of 500 V, duration 0.2 s. Electrical impulses were provided by a certified rodent shocker (Hugo Sachs Elektronik, Freiburg, Germany). 

### 4.4. Isobolographic Analysis

Since ranolazine behaves like an anticonvulsant drug in the MES test and its ED_50_ value was determinable, we used isobolographic analysis to characterize its pharmacological interactions with antiepileptic drugs. To perform this analysis, ranolazine was combined with each antiepileptic drug in three fixed-dose ratios of 1:3, 1:1, and 3:1. Then, the ED_50mix_ values were determined experimentally as the total effective doses of two-drug mixtures that actually protected 50% of animals against electroconvulsions. Simultaneously, the ED_50add_ values were evaluated for each drug combination. The ED_50add_ reflected the total additive dose of the two drugs in the mixture calculated from the line of additivity that theoretically protected 50% of mice against electroconvulsions. Both experimental ED_50mix_ and theoretical ED_50add_ values were determined from the dose–response relationships of two-drug combinations according to Litchfield and Wilcoxon [31] and Tallarida [32]. The calculated 95% confidence limits of ED_50_ values were then transformed to standard errors of the means (SEMs). Experimental ED_50mix_s and theoretical ED_50add_s were statistically compared by use of the unpaired Student’s *t* test according to Porecca et al. [33] and Tallarida [32]. When the difference is not statistically significant, the interaction is regarded as additive. If the ED_50mix_ is significantly lower than the respective ED_50add_, the interaction is classified as synergistic. In contrast, when the ED_50mix_ is significantly greater than the corresponding ED_50add_, the interaction is considered to be antagonistic. Detailed information on the methodology of isobolographic analysis of two-drug interactions can be found in previous studies [34,35].

### 4.5. Chimney Test

The chimney test is used to evaluate motor coordination in mice [36]. The chimney is a 25 cm long plastic cylinder with inner diameter of 3 cm and internal threading facilitating the movement of animals. On the first day, untreated mice from control and experimental groups were inserted separately at the beginning of the horizontally positioned tube. When the animal reached the end of the cylinder, it was placed vertically. Thus, the mouse had to get out of the chimney backwards. Animals that had not left the cylinder within 60 s were excluded from further investigations. Twenty-four hours later, the same mice were applied with vehicle or combinations of ranolazine with antiepileptic drugs at two dose-fixed ratios of 1:3 and 1:1, at which antagonism was observed in the MES test. Then, the animals performed the chimney test for the second time. Those that failed the test within 60 s were considered to have impaired motor coordination. 

### 4.6. Measurement of Brain Concentrations of Antiepileptic Drugs

Brain concentrations of valproate, carbamazepine, phenytoin, and phenobarbital were measured to verify possible pharmacokinetic interactions between ranolazine and listed antiepileptic drugs. Control animals were applied with vehicle and a respective antiepileptic drug. Animals from investigated groups were administered with combinations of ranolazine with antiepileptics in the proportions of 1:3 and 1:1. In the two proportions, antagonistic interactions were found in the isobolographic analysis. At times used in behavioral tests, mice were decapitated, and their brains removed from the skulls. The brains were homogenized by Ultra Turax T8 homogenizer (IKA, Staufen, Germany) with Abbott buffer (2:1 vol/weight) and centrifuged at 10,000× *g* for 10 min. Supernatants obtained (75 µL) were used for drug concentration analysis by fluorescence polarization immunoassay (Architect c4000 analyzer, Abbott Laboratories, Warsaw, Poland). Results were expressed presented as means ± SD (µg/mL). 

### 4.7. Statistics

The ED_50mix_ values with 95% confidence limits were calculated in the Litchfield and Wilcoxon log-probit evaluation [32]. Confident limits were then transformed to standard errors (SEMs). Multiple comparisons of the ED_50mix_ and ED_50add_ values (± SEM) were performed by the one-way analysis of variance (ANOVA) with the post hoc Tukey test. 

Results obtained in the chimney test were analyzed by the Fisher’s exact probability test for qualitative variables. Statistical analysis of differences in brain concentrations of antiepileptic drugs was carried out by the unpaired Student’s *t* test. The level of significance was established as *p* < 0.05.

## 5. Conclusions

As far as our experimental findings can be extrapolated into clinical conditions, antagonism determined between ranolazine and carbamazepine, phenytoin, or phenobarbital may suggest careful use of ranolazine as an antiarrhythmic medication in patients with epilepsy. Although the delayed lethality induced by ranolazine is not reflected in clinical trials, mechanisms of this phenomenon occurring in mice require detailed explanation. 

## Data Availability

The data presented in this study are available in the article.

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
