# Peer review of "Ranolazine Interacts Antagonistically with Some Classical Antiepileptic Drugs—An Isobolographic Analysis"

_molecules, 2022, doi:10.3390/molecules27248955_

Round 1

Reviewer 1 Report

There are some spelling errors in the manuscript. Moreover, on page 5 line 176 there is a mistake regarding pharmacokinetic interactions - it should be: Remaining interactions seem to be pharmacodynamic.

I would suggest to the authors to alter the conclusion, especially in the abstract. All experiments were performed in mice and the results cannot be extrapolated to humans. The fact that ranolazine was lethal in mice does not mean it would have any adverse effects in humans, specially given the discussed fact that it is used as an antiarrhytmic in doses up to 2000 mg/day.

Author Response

Response to Reviewer 1

Thank you very much for valuable comments. I hope that my answers will be satisfactory.

Comments and answers:

  1. There are some spelling errors in the manuscript. Moreover, on page 5 line 176 there is a mistake regarding pharmacokinetic interactions - it should be: Remaining interactions seem to be pharmacodynamic.

This mistake has been corrected. Also, the entire manuscript was reviewed for typing errors, all of them have been corrected.

  1. I would suggest to the authors to alter the conclusion, especially in the abstract. All experiments were performed in mice and the results cannot be extrapolated to humans. The fact that ranolazine was lethal in mice does not mean it would have any adverse effects in humans, especially given the discussed fact that it is used as an antiarrhytmic in doses up to 2000 mg/day.

The conclusion in the Abstract has been reformulated to: … ranolazine exhibits clear-cut anticonvulsant properties in the MES test but may interact antagonistically with some antiepileptic drugs. The obtained results need confirmation in clinical studies.

In the Conclusion section, the first sentence has been reformulated and phrase “may suggest careful use of ranolazine as an antiarrhythmic medication in patients with epilepsy” has been included.

Reviewer 2 Report

This interesting work describes the effect of ranolazine, an antianginal and antiarrhytmic drug, on maximal electroshock-induced seizures in mice as well as interactions between ranolazine and classical antiepileptic drugs (valproate, carbamazepine, phenytoin and phenobarbital ) in this model of epilepsy. From a clinical point of view, this is important issue and useful research, for epilepsy sufferers who need to take antiarrhythmic medications.

In general, the paper is well written and I recommend it for publication, however, it requires some corrections.

Minor remarks:

1. In introduction, there is no explanation why the MES test was used as an experimental model of epilepsy.

2. Fig.1 should be corrected because it is not clear.

Author Response

Response to Reviewer 2

I would like to express my gratitude for the review. Below there are my answers to all remarks:

  1. In introduction, there is no explanation why the MES test was used as an experimental model of epilepsy.

Such an explanation has been added in the end of the Introduction section: “Maximal electroshock is a standard screening method for the identification of potential anticonvulsant drugs. It has been used for years due to the ease and low costs of performing. For the same reasons, this model is attractive for carrying out isobolographic analysis of drug interactions”.

  1. Fig.1 should be corrected because it is not clear.

Fig. 2 has been rewritten, now it seems much clearer. I hope, it is satisfactory at present. If not, I can try to divide this figure into four separate ones.

Reviewer 3 Report

The manuscript of Borowicz-Reutt and Banach entitled “Ranolazine interacts antagonistically with some classical antiepileptic drugs – an isobolographic analysis” deals with interactions between ranolazine and conventional epileptic drugs in the maximal electroshock in mice. Ranolazine presented properties of anticonvulsant drugs in this test but interacted antagonistically with three of four tested antiepileptic drugs. Results of this study seem to be interesting primarily due to the wide use of this drug in cardiology. Surprisingly, the antianginal drug, quite safe in humans, occurred toxic in mice. This is a very interesting topic. Nevertheless, there are several comments regarding the submitted manuscript:

My major comment:

1.      The authors mentioned that mechanisms of action of ranolazine resembles those of lidocaine, mexiletine, and amiodarone. I suggest extending the discussion to interactions between these drugs and antiepileptics, as long as they are known.

Minor points:

1.      Abstract: in the sentence “Ranolazine and its combinations with …. ranolazine.. “, the second ranolazine should be replaced with phenobarbital.

2.      In Introduction (Pg. 2):

-        sentences beginning with “Mutations in SCN1A …” and “Nevertheless, the drug …” are not clear enough and should be reformulated.

-        in the sentence beginning with “The most common …”, the word ‘are’ should be replaced with ‘were’.

3.      In the last sentence of Introduction, the model of seizures in which the experiments were performed should be added.

4.      The legend to Table 3 – the fragment about statistical analysis of results from the passive avoidance test should be deleted, since such results were not presented.

5.      Results 2.3:

-        The first sentence is not clear enough and should be reformulated

-        In the next sentence, the fragment “of drug combination” should be eliminated

-        In the further text, ‘ranolasine’ should be replaced with ‘ranolazine’

-        Figure 2 should be transferred from Discussion to the end of the point 2.3 of the Results

6.      Discussion:

-        The two first sentences contain overlapping information. Please, reformulate them.

-        Pg.6 below Fig. 1: the doubled fragment “phenytoin inhibits not only INaF, but also INaL currents [7].” Should be deleted

-        Pg.7: to the sentence beginning with “According to authors …” should be supplied with the fragment ‘treated by these drugs’.

-        Pg.7: instead of “in proportions leading to antagonistic interactions”, specific proportions should be listed  

-        Pg.7: ‘up to 6 of 10 mice’ sounds better than “up to 6 mice”

-        The last sentence in Discussion should be developed. Why, in the opinion of Authors,  inhibition of IKr currents may contribute to the ranolazine-induced lethality?

Author Response

Thank you for all comments and remarks, hoping that my answers will occur satisfactory:

Comments and answers:

  1. 1.The authors mentioned that mechanisms of action of ranolazine resembles those of lidocaine, mexiletine, and amiodarone. I suggest extending the discussion to interactions between these drugs and antiepileptics, as long as they are known.

Appropriate fragment has been added on page 7, beginning with “As it was mentioned…”.

Minor points:

  1. Abstract: in the sentence “Ranolazine and its combinations with …. ranolazine.. “, the second ranolazine should be replaced with phenobarbital.

The error has been corrected

  1. 2.In Introduction (Pg. 2):

-        sentences beginning with “Mutations in SCN1A …” and “Nevertheless, the drug …” are not clear enough and should be reformulated.

The first sentence has been divided in two ones as follows: Mutations in SCN1A gene coding the alpha subunit of NaV1.1 are associated with generalized epilepsy with febrile seizures, Dravet syndrome (severe myoclonic epilepsy in infancy), and inherited familial migraine. Simultaneously, such mutations resulted in an increase in persistent sodium currents [7].

The second sentence has been reformulated to the following version: However, the drug did not significantly affect the voltage-gated potassium channels or NMDA- and GABA-related neurotransmissions. Also, the neurotransmitter release to the synaptic cleft was not influenced [8].

-        in the sentence beginning with “The most common …”, the word ‘are’ should be replaced with ‘were’.

This word has been replaced.

  1. In the last sentence of Introduction, the model of seizures in which the experiments were performed should be added.

The maximal electroshock test in mice has been added to this sentence. Additionally, the choice of this test has been elucidated.

  1. The legend to Table 3 – the fragment about statistical analysis of results from the passive avoidance test should be deleted, since such results were not presented.

This fragment has been deleted.

  1. Results 2.3:

-        The first sentence is not clear enough and should be reformulated

The sentence was reformulated to: Ranolazine was combined with antiepileptic drugs in two proportions (1:3 and 1:1), in which antagonism was observed between the antianginal drug and carbamazepine, phenytoin and phenobarbital. For valproate, the same proportions were used to maintain comparability of results.

-        In the next sentence, the fragment “of drug combination” should be eliminated

This fragment has been deleted.

-        In the further text, ‘ranolasine’ should be replaced with ‘ranolazine’

This error has been corrected.

-        Figure 2 should be transferred from Discussion to the end of the point 2.3 of the Results

Figure 2 has been transferred, however, it had to be diminished to fit it to the Pg.5.

  1. Discussion:

-        The two first sentences contain overlapping information. Please, reformulate them.

At present, this sentence looks like: Results of the present study revealed that ranolazine behaved like a regular antiepileptic drug in the maximal electroshock test in mice.

-        Pg.6 below Fig. 1: the doubled fragment “phenytoin inhibits not only INaF, but also INaL currents [7].” Should be deleted

This fragment has been deleted.

-        Pg.7: to the sentence beginning with “According to authors …” should be supplied with the fragment ‘treated by these drugs’.

This fragment has been added.

-        Pg.7: instead of “in proportions leading to antagonistic interactions”, specific proportions should be listed  

This change has been introduced.

-        Pg.7: ‘up to 6 of 10 mice’ sounds better than “up to 6 mice”

This fragment has been reformulated.

-        The last sentence in Discussion should be developed. Why, in the opinion of Authors,  inhibition of IKr currents may contribute to the ranolazine-induced lethality?

This sentence has been widened to: It can only be suggested that ranolazine-related inhibition of IKr and IKs currents may contribute to this phenomenon in the mechanism of inducing severe arrhythmias and/or seizures.

Round 2

Reviewer 3 Report

I recommend the article for publication in the revised form.

Author Response

Thank you for approving the article in the revised form.

Sincerely yours,

Kinga Borowicz-Reutt